# Bereaved Family Members’ Perspectives of Good Death and Quality of End-of-Life Care for Malignant Pleural Mesothelioma Patients: A Cross-Sectional Study

**DOI:** 10.3390/jcm11092541

**Published:** 2022-05-01

**Authors:** Yasuko Nagamatsu, Yumi Sakyo, Edward Barroga, Riwa Koni, Yuji Natori, Mitsunori Miyashita

**Affiliations:** 1Graduate School of Nursing Science, St. Luke’s International University, 10-1 Akashi-cho, Chuo-ku, Tokyo 104-0044, Japan; yumi-sakyo@slcn.ac.jp (Y.S.); edward-barroga@slcn.ac.jp (E.B.); 2St. Luke’s International Hospital, 9-1 Akashi-cho, Chuo-ku, Tokyo 104-8560, Japan; riwakoni@luke.ac.jp; 3Hirano Kameido Himawari Clinic, 7-10-1 Kameido, Koto-ku, Tokyo 136-0071, Japan; natori@himawari-clinic.jp; 4Department of Palliative Nursing, Health Sciences, Tohoku University Graduate School of Medicine, 2-1 Seiryomachi, Aoba-ku, Sendai 980-8575, Japan; miya@med.tohoku.ac.jp

**Keywords:** mesothelioma, asbestos, rare lung disease, palliative care, good death, quality of care

## Abstract

Objective: This study investigated whether malignant pleural mesothelioma (MPM) patients achieved good deaths and good quality of end-of-life care compared with other cancer patients from the perspective of bereaved family members in Japan. Methods: This cross-sectional study was part of a larger study on the achievement of good deaths of MPM patients and the bereavement of their family members. Bereaved family members of MPM patients in Japan (*n* = 72) were surveyed. The Good Death Inventory (GDI) was used to assess the achievement of good death. The short version of the Care Evaluation Scale (CES) version 2 was used to assess the quality of end-of-life care. The GDI and CES scores of MPM patients were compared with those of a Japanese cancer population from a previous study. Results: MPM patients failed to achieve good deaths. Only 12.5% of the MPM patients were free from physical pain. The GDI scores of most of the MPM patients were significantly lower than those of the Japanese cancer population. The CES scores indicated a significantly poorer quality of end-of-life care for the MPM patients than the Japanese cancer population. The total GDI and CES scores were correlated (r = 0.55). Conclusions: The quality of end-of-life care for MPM patients remains poor. Moreover, MPM patients do not achieve good deaths from the perspective of their bereaved family members.

## 1. Introduction

Malignant pleural mesothelioma (MPM) is a rare fatal malignancy caused mainly by asbestos [1]. The number of people with MPM who die each year in Japan is about 1550, and that number is growing [2]. It is estimated that Japan will have 66,000–100,000 deaths from mesothelioma between the years 2003 and 2050 [3,4]. The median survival from the time of diagnosis in Japan is 7.9 months [5]. MPM causes a series of debilitating physical symptoms, such as chest pain, dyspnea, fatigue, anorexia, insomnia, constipation, and sweating [6,7,8,9,10,11]. Psychological issues, such as uncertainty, lack of control [12], memory problems, difficulties in concentrating, feeling that problems cannot be solved [13], depression, anxiety, fear, and isolation [8], all negatively affect the quality of life of MPM patients. Finally, there is additional psychological distress for victims of the asbestos industry [14]. Suffering from asbestos-related disease causes fear of premature death [15]. MPM patients in Japan reportedly suffer from physical and psychological distress [16], and their quality of life is impaired [9].

Lamentably, the quality of life of MPM patients in the terminal stage, particularly their achievements of good deaths and good quality of end-of-life care, has been scarcely researched and thus remains poorly understood. Unfortunately, there are barriers to conducting research on MPM patients in their terminal stage. These include their small population, and the short lengths of time between disease diagnosis, debilitation, and death. Moreover, conducting research on terminally ill patients imposes unnecessary burdens on them. Therefore, many studies are conducted with bereaved family members [13,17,18,19,20] to evaluate the patients’ achievements of good deaths and the quality of their end-of-life care.

This study aimed to investigate whether MPM patients achieved good deaths and good quality of end-of-life care compared with other cancer patients in Japan from the perspective of bereaved family members. The data for the other cancer patients in Japan were taken from a previous study [21].

## 2. Methods

### 2.1. Study Design, Participants, and Setting

This study used a cross-sectional survey design to examine the achievements of good death and good quality of end-of-life care for MPM patients from the perspective of bereaved family members.

The inclusion criteria for bereaved family members were as follows: (1) had lost a loved one to MPM, (2) had a loved one who had been diagnosed with MPM after 2008 when the first evidence-based chemotherapy succeeded in prolonging the survival of MPM patients, and (3) could respond to a self-administered questionnaire written in Japanese. The exclusion criterion was a bereaved family member who had experienced a loss within six months. This research is part of a larger study which also investigated the complicated grief of the bereaved family members of MPM patients. According to the previous study, the diagnosis of complicated grief should be made at least six months after the death of a family member [22].

A request for cooperation was sent to the advocacy group of the Japan Association of Mesothelioma and Asbestos-Related Disease Victims and their Families. The association has 15 branches across Japan and works with approximately 700 victims of asbestos-related diseases and their families. The association sent the informed consent information and questionnaires to 109 bereaved family members in November 2016. Those agreeing to participate returned the completed questionnaires via postal mail by March 2017.

### 2.2. Outcomes

The primary outcomes were the achievements of a good death and good quality of end-of-life care for MPM patients. The secondary outcome was the presence of the common symptoms of MPM.

### 2.3. Instruments

#### 2.3.1. Information of Patients and Bereaved Family Members

The following information was provided by the bereaved family members about the deceased patients: sex, age at diagnosis, survival and received treatments, receipt of two types of insurance compensation benefits, and place of death.

The information about the bereaved family members included the following: age, relationship to the patient, time of bereavement, experience of end-of-life discussion with the patient, timing of patient’s death, financial impact of patient’s MPM on family, and level of anger toward asbestos. The bereaved family members were also asked about their satisfaction with care on diagnosis, when the patient became critical, and when the patient died.

#### 2.3.2. Good Death Inventory

Achievement of good death was measured using the Good Death Inventory (GDI), which had internal consistency (α = 0.74–0.95) and acceptable test–retest reliability (intra-class correlation coefficient = 0.38–0.72) [17]. The GDI was validated to evaluate the achievement of good death from the perspective of bereaved family members in Japan [17]. The GDI has 18 items consisting of 10 core items and 8 optional items, and is answered using a seven-point Likert scale (1 = absolutely disagree, 7 = absolutely agree). The possible scores range from 18 to 126, and a high score indicates the achievement of a good death.

#### 2.3.3. Care Evaluation Scale

The short version of the Care Evaluation Scale (CES) version 2 (Cronbach’s α = 0.96) was used to evaluate the quality of end-of-life care in Japan [23]. The CES consists of 10 items. The bereaved family members answered using a six-point Likert scale (1 = highly disagree, 6 = highly agree). A high total CES score indicates a good quality of end-of-life care.

#### 2.3.4. Symptoms

The presence of the common symptoms of MPM, namely, pain, dyspnea, anorexia, fatigue, anxiety, dysphagia, constipation, nausea, insomnia, edema, and palpitation, was asked with respect to two time points. These time points were (1) at the end of chemotherapy (only for the bereaved family members of patients who received chemotherapy—i.e., when chemotherapy was stopped, being no longer effective), and (2) at the final critical stage (i.e., when the patient entered the critical stage). The bereaved family members checked the items of symptoms the MPM patients experienced. These two time points enabled the comparison of the present results with previous results that reported on the care needs of patients because of their severe symptoms [16].

### 2.4. Missing Data

Mean imputation was conducted for the missing data of GDI and CES scores according to the instructions for the tools.

### 2.5. Comparison of Study Data

A nationwide project to evaluate hospice and palliative care in Japan was previously conducted by Miyashita et al. and reported as the Japan Hospice and Palliative care Evaluation (J-HOPE) study [21]. This project evaluated the end-of-life care of cancer patients from the perspective of bereaved family members in nationwide designated cancer centers, inpatient palliative care units (PCUs), and home hospices. The study focused on care satisfaction, the structure and process of care, and the achievement of a good death. This previous study compared the data according to the last place of care. Data from this previous study were provided to us by Dr. Miyashita, who is a co-author of the present study. There were 8398 questionnaire responses from family members that were analyzed by Miyashita et al. [24].

### 2.6. Statistical Analysis

The scores of each scale were calculated using a previously reported scoring procedure [17,23]. The scores of the measurement tool items in GDI and CES were totaled and compared with those of cancer patients in the J-HOPE study [21]. The GDI and CES mean scores in the J-HOPE study [21] were calculated according to the place of death and compared with the GDI and CES mean scores in the present study. The achievements of good death (measured using GDI) and good quality of end-of-life care (measured using CES) scores in the present MPM study and the previous J-HOPE study were compared using the binominal test. The GDI and CES total scores in the present MPM study and the previous J-HOPE study were compared using a one-sample *t*-test.

The correlations between the GDI and the CES were examined. Thereafter, the GDI scores and the patients’ and bereaved participants’ information were examined. Sex, receiving treatments, approval for compensation, experience of end-of-life discussion with patients, and satisfaction of care were treated as dichotomous variables. Finally, the coefficients and their 95% confidence intervals estimated by multiple regression analysis were used to assess the correlations between the GDI and CES scores and the clinical social factors. A *p*-value of < 0.05 was considered to indicate a statistically significant difference. Statistical analysis was performed using SPSS version 27.

### 2.7. Ethical Consideration

This study was approved by the Research Ethics Committee of St. Luke’s International University (16-A035). It was conducted based on the ethical principles of avoiding harm, voluntary participation, anonymity, and the protection of privacy and personal information.

## 3. Results

Of the 109 questionnaires distributed to the bereaved family members through the related victims and family advocacy group, 74 (67.9%) were completed and returned via postal mail by the end of March 2017. Two bereaved family member respondents who had experienced a loss within the last six months were excluded. Thus, a total of 72 questionnaires were analyzed.

### 3.1. Characteristics of Malignant Pleural Mesothelioma Patients and Bereaved Family Members

As shown in Table 1, 81.9% of the deceased MPM patients were men, and their mean age at diagnosis was 66.9 years. The treatment modalities they received were chemotherapy (70.8%), palliative care (56.9%), and surgery (19.4%). A large minority (48.6%) died in the respiratory ward, followed by the PCU or hospice (33.3%). Only 13.9% died at home. The mean survival time was 14.5 months from the time of diagnosis. The majority of the bereaved family members (72.2%) was spouses of the MPM patients, and the mean bereavement time was 45.2 months.

### 3.2. Achievement of Good Death

The obtained data revealed that MPM patients failed to achieve good deaths. The mean total GDI score of the MPM patients was 61.9 ± 15.7, which was significantly lower than the 81.1 of the J-HOPE cancer patients. Figure 1 shows the comparison of the percentage scores of MPM patients and J-HOPE cancer patients for the GDI items for the achievement of good death. The lowest percentages of achievement by the MPM patients in the 10 core items of the GDI were for the items “being free from physical distress” (12.5%) followed by “feeling that life is completed” (18.1%) and “having some pleasure in daily life” (27.8%). The binominal test showed that the percentages regarding the achievement of a good death in the MPM patients were significantly lower than those in the J-HOPE cancer patients in all items, except for the following four items: “being independent in daily activities”, “knowing what to expect about the future condition”, “living in calm circumstances”, and “supported by religion”. The greatest gaps in the achievement of good death between the MPM patients and the J-HOPE cancer patients were for “being free from physical distress”, which was true for 12.5% of the MPM patients compared with 64.7% of the J-HOPE cancer patients, followed by “not exposing one’s physical and mental weakness to family”, “dying a natural death”, and “feeling life is completed”.

### 3.3. Quality of End-of-Life Care

The total scores of CES in the MPM patients and the J-HOPE cancer patients were significantly different, as shown in Figure 2. The mean total score of CES in the MPM patients was 70.3 ± 16.0, which was significantly lower than the 75.8 in the J-HOPE cancer patients. The binominal test showed that all the scores of the CES items indicated a significantly poorer quality of end-of-life care in the MPM patients than in the J-HOPE cancer patients except in the items “cost”, “coordination and consistency”, and “explanation to family by physician”.

### 3.4. Symptoms

The percentages of MPM patients who experienced symptoms at the end of chemotherapy are shown in Figure 3, and the same percentages at the final critical stage are shown in Figure 4. More than half of the MPM patients experienced pain, dyspnea, anorexia, and anxiety at the end of chemotherapy. When the MPM patients reached the final critical stage, symptoms such as fatigue and dysphasia followed.

### 3.5. Factors Associated with a Good Death

The GDI and CES total scores were significantly associated (correlation coefficient *ρ* = 0.554, *p* = 0.0001), indicating that the patients who received better end-of-life care were more likely to achieve good deaths. The multiple regression analysis results are shown in Table 2.

The final regression model for predicting good death showed that higher GDI scores were significantly related to the surveyed family member being female, the patient dying later than expected, and satisfaction with care when the patient became critical.

### 3.6. Factors Associated with Quality of End-of-Life Care

The final regression model for predicting good death (Table 3) showed that higher CES scores were significantly related to the following: satisfied with the care received when the patient died, and Received chemotherapy.

## 4. Discussion

In this study, we described the extent to which Japanese MPM patients achieved good deaths and their good quality of end-of-life care. The findings were compared with those of a large cohort of Japanese cancer patients from the J-HOPE study [21].

The present results demonstrate a lack of good deaths among MPM patients. The three main findings of this study are as follows: (1) there was a remarkable lack of good deaths among the MPM patients; (2) there was an enormous burden of symptoms in the MPM patients; and (3) the quality of end-of-life care in the MPM patients was poorer than that in the J-HOPE cancer patients. The CES score was correlated with the GDI score, consistent with the findings of Miyashita et al. [17]. The final regression model showed that a higher GDI score was significantly related to the surveyed family member being female, the patient dying later than expected, and satisfaction with the care received when the MPM patient became critical.

### 4.1. Poor Achievement of Good Death

This study showed an extreme lack of good deaths among the MPM patients. The lowest score from among the 10 GDI core items was for the item “being free from physical distress” (12.5%), which was significantly lower than the 62.9% score for the Japanese cancer population [21]. Symptom management is difficult in MPM patients, possibly because (1) MPM progresses rapidly and causes a variety of severe symptoms [6,9,25,26]; and (2) MPM results in anger and negative feelings of injustice [7,14,16], which tend to complicate the patient’s physiological distress more than other malignancies. Additionally, MPM has the potential to cause spiritual pain. Some studies have advocated care to ease the spiritual pain of MPM patients [27,28].

Only 18.1% of the MPM patients in the present study had the “feeling that life is completed”, which was significantly lower than the figure of 49.9% among the cancer population [21]. The possible reasons are as follows: (1) In this current study, the mean age of diagnosis was 66.9 years, and the mean survival time was only 14.5 months. The patients died relatively young, and they had very little time to complete their lives and face their deaths. (2) As the cause of MPM was asbestos and not one’s own doing, the patient may have felt that death from MPM was unfair.

For patients with MPM, “Dying without awareness that one is dying” (4.2%) was, for the most part, not possible. Patients were told at the time of their diagnosis that their disease was incurable [7].

Only 11.1% of the MPM patients felt “supported by religion”; however, this percentage was not significantly different from the 19.6% of the cancer population [21]. As Ando et al. [29] reported, religious care is not very common in Japan.

The multiple regression analysis showed that the family member surveyed being female, the patient dying later than expected, and satisfaction with care when the patient became critically ill were related to the GDI score. It is not clear why the family member surveyed being female was related to a higher GDI score. One possibility is that a higher number of Japanese females do not work and focus on caregiving; however, we did not ask about the jobs of the bereaved family members. It is necessary to investigate the relationship between the gender of the family member and the achievement of a good death. Carr [30] reported that the interval between the onset of terminal illness and death provided opportunities for people to plan their end-of-life care. However, an MPM diagnosis leaves a much shorter time for patients than in most cases, especially for those who died sooner than expected, reducing their capacity to prepare for good deaths.

The satisfaction with care when patients become critical is related to the achievement of a good death, which is consistent with the findings in the “Good Death” study by Miyashita et al. [17]. For patients with MPM to achieve a good death, preparation for the acute exacerbation of the disease and the implementation of physical, psychological, and spiritual care in a timely manner are crucial.

### 4.2. Heavy Symptom Burden

The present results show that the MPM patients experienced various kinds of symptoms. As shown in other published studies [6,9,25,26,31], pain, dyspnea, anorexia, and fatigue were the major symptoms exhibited by the MPM patients. The major symptoms of MPM patients are similar to the major symptoms of lung cancer patients, with a high prevalence of pain, fatigue, dyspnea, anorexia, and anxiety [6,32]. An important outcome of the present study was that it revealed the high prevalence of the various symptoms of MPM patients at the end of chemotherapy. For symptom management in MPM, several studies have recommended the introduction of palliative care in the early stages of MPM [26,33]. Unfortunately, similarly to cancer patients [34], MPM patients often refuse palliative care because of their denial of the fatal nature of the disease. They are thus unwilling to end their anticancer treatment and enter palliative care. Advanced care planning is encouraged; however, this is challenging for MPM patients, who have short prognoses. Horne et al. reported that discussions about end-of-life care planning following the disclosure of a terminal prognosis caused a feeling of abandonment [35].

### 4.3. Poor Quality of End-of-Life Care

The present results show a poor quality of end-of-life care for MPM patients in Japan and significantly worse care than for other cancer patients. The possible reasons for this poor quality of end-of-life care could be (1) the limited availability of treatment for MPM, which has recently improved in Japan [36]; and (2) the health providers’ lack of knowledge and skills regarding the treatment and care of MPM patients [8]. As the multiple regression analysis showed that “Satisfaction with the care received when the patient died” and “Received chemotherapy” were related to the CES score, improvements in end-of-life care are recommended through (1) the assurance of quality care on the death bed, and (2) the provision of continuous end-of-life care to patients who do not receive chemotherapy.

### 4.4. Implications for Care and Further Research

The MPM patients experienced various symptoms at the end of chemotherapy and when they entered the final critical stage. Medical professionals need to understand that MPM patients develop various symptoms in the early stages of the disease, even when treated with chemotherapy. Thus, medical professionals need to inform MPM patients regarding the possible symptoms that they will encounter and advise them on how to prepare, which may be challenging for patients. To support MPM patients at this difficult time, transition care is crucial. The care for MPM patients must include (1) symptom management from the earliest stage; (2) care for psychological, social, and spiritual pain; and (3) care for their families as provided by a multidisciplinary team, consisting of a patient and family advocacy group, and a lawyer [10,27,28].

### 4.5. Limitations

This study has some limitations. First, not all of the bereaved family members of the deceased MPM patients were contacted, as Japan has no registration system for MPM patients. Therefore, this study had a small sample. Second, as the participants were members of the advocacy group, it is uncertain whether the results are representative of the general population of bereaved family members of deceased MPM patients. The patients and family advocacy group, with their network of medical staff and hospitals, may have represented bereaved family members who are less distressed by the care their loved ones receive, thus representing a biased group. Third, the mean number of months of bereavement was 45.2; therefore, the participants may have had recall bias or forgotten key factors. Finally, this study was a cross-sectional study, and therefore, no causal relationships were established. To overcome the limitations regarding representativeness, it is necessary to conduct census surveys based on an MPM registration system, as this will allow representative random samplings.

## 5. Conclusions

This cross-sectional study revealed the remarkably rare achievement of a good death among MPM patients in Japan. The MPM patients experienced an enormous burden from their symptoms and were seldom free of physical distress. Another challenge faced by MPM patients in the achievement of a good death was the sense of life completion, which was difficult for patients with MPM caused by asbestos. The quality of end-of-life care of MPM patients was poorer than that of other cancer patients. The GDI score of the MPM patients was closely correlated with their CES score. Further research and interventions are urgently required, aimed at achieving a good death for MPM patients by providing quality continuous care, including (1) symptom management from the earliest stage; (2) care for psychological, social, and spiritual pain; and (3) care for their families as provided by a multidisciplinary team.

## Figures and Tables

**Figure 1 jcm-11-02541-f001:**
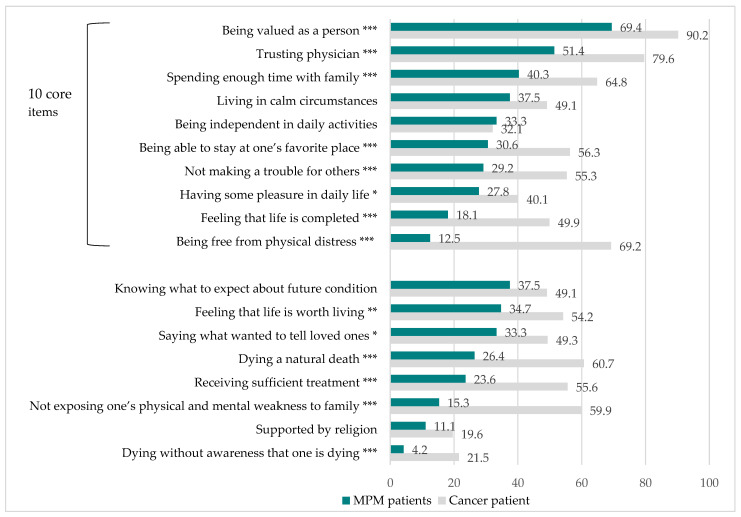
Comparison of the percentages of MPM patients and J-HOPE cancer patients concerning GDI items for the achievement of a good death. Sum of “somewhat agree”, “agree”, and “absolutely agree” responses. Data of cancer patients were from the J-HOPE national survey of Japanese cancer patients (reference [21]). Weighted means of GDI scores in general cancer patients in Japan (reference [21]) were calculated according to the place of death. Core and optional items were established by factor analysis (reference [17]). * *p* < 0.05, ** *p* < 0.005, *** *p* < 0.001.

**Figure 2 jcm-11-02541-f002:**
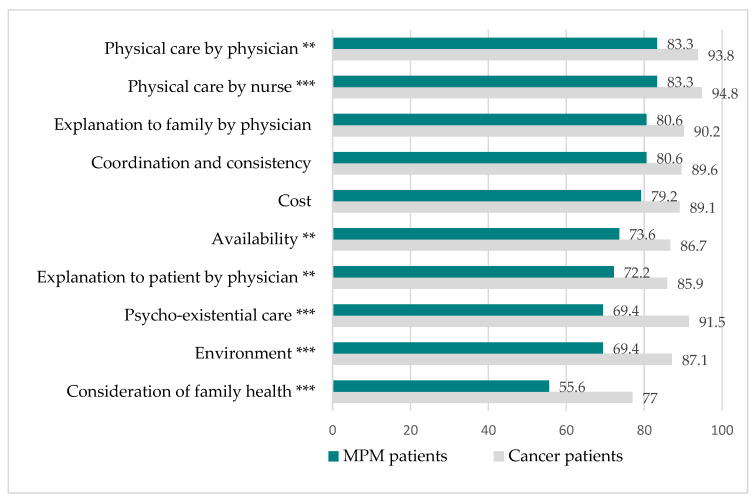
Comparison of the percentages of MPM and J-HOPE cancer patients with regard to CES items for achieving good quality end-of-life care. Sum of “somewhat agree”, “agree”, and “absolutely agree”. The weighted means of CES scores in general cancer patients in Japan were calculated according to the place of death. Data are from the J-HOPE study (Reference [21]). ** *p* < 0.005, *** *p* < 0.001.

**Figure 3 jcm-11-02541-f003:**
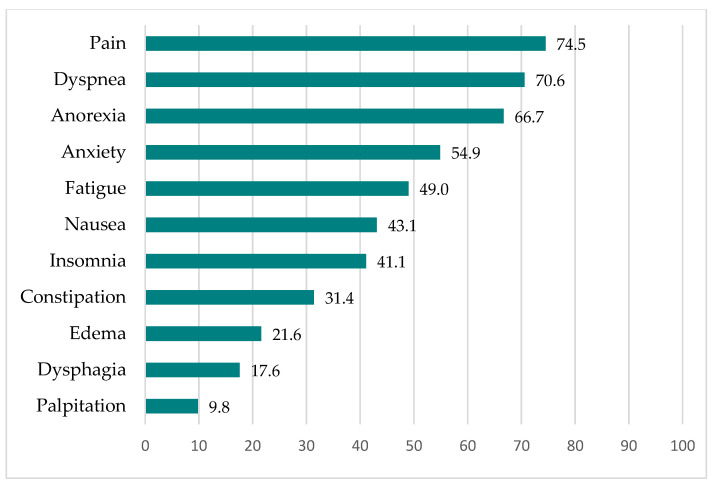
Percentages of MPM patients experiencing symptoms at the end of chemotherapy (*n* = 51).

**Figure 4 jcm-11-02541-f004:**
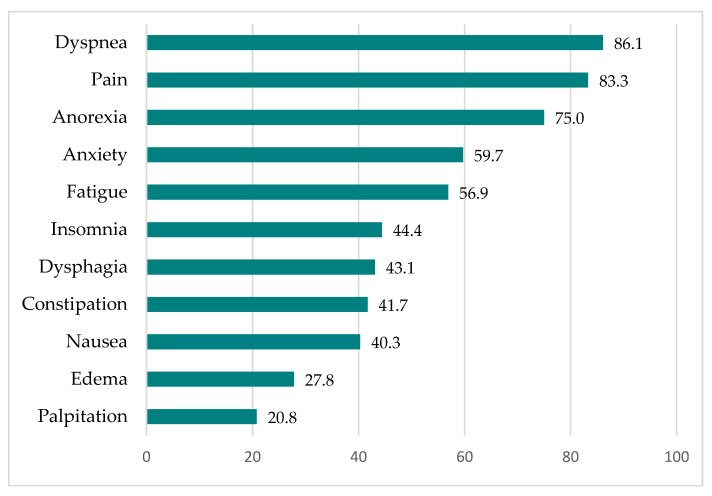
Percentages of MPM patients experiencing symptoms at the final critical stage (*n* = 72).

**Table 1 jcm-11-02541-t001:** Comparison of the characteristics of malignant pleural mesothelioma patients and cancer patients, and their bereaved participants.

Disease				MPM		Cancer *					
				*n* = 72		Place of Death					
						Designated Cancer Center (*n* = 2794)		Palliative Care Unit (*n* = 5312)		Home Hospice (*n* = 292)	
Patients				*n*	%	*n*	%	*n*	%	*n*	%
Sex	Men			59	81.9	1820	65.1	2906	54.7	181	62
	Women			13	18.1	973	34.8	2364	44.5	111	38
Primary cancer site	Pleura **			72	100	-	-	-	-	-	-
	Lung			0	0	688	24.6	1246	23.5	63	21.6
	Stomach			0	0	395	14.1	635	12	36	12.3
	Colorectum/rectum			0	0	260	9.3	651	12.3	54	18.5
	Liver			0	0	279	10	281	5.3	18	6.2
	Gall bladder/bile duct			0	0	165	5.9	201	3.8	14	4.8
	Pancreas			0	0	243	8.7	398	7.5	18	6.2
	Esophagus			0	0	112	4	184	3.5	8	2.7
	Breast			0	0	83	3	266	5	8	2.7
	Others			-	-	513	18.4	1389	26.2	69	23.7
Source of asbestos exposure	Occupation			49	68.1						
	Neighboring factory			17	23.6						
	School			1	1.4						
	Family			1	1.4						
	Unknown			4	5.4						
Treatment	Surgery			14	19.4						
(includes multiple treatments)		Extrapleural pneumonectomy		12	16.7						
		Pleurectomy decoration		2	2.8						
	Chemotherapy			51	70.8						
	Radiotherapy			15	20.8						
	Palliative care			41	56.9						
Compensated	Workmen’s accident compensation insurance			47	65.3						
(some had both types)	Asbestos-related health damage relief system			56	77.8						
Place of death	Respiratory ward			35	48.6						
	Palliative care unit/hospice			24	33.3						
	Home			10	13.9						
	Other			3	4.2						
Age at diagnosis (years)	Range:		36–92	Mean ± SD	66.9 ± 9.6	69.8 ± 11.5		70.9 ± 12.1		71.8 ± 13.0	
Survival (months)			0.5–69		14.5 ± 14.1						
**Bereaved family members**				*n*	%	*n*	%	*n*	%	*n*	%
Sex	Men			15	20.8	825	29.5	1694	31.9	60	20.6
	Women			57	79.2	1696	60.7	3556	67.1	228	78.1
Relationship with patient	Spouse			52	72.2	1535	54.9	2506	47.2	165	56.5
	Child			20	17.8	672	24.1	1809	34.1	78	26.7
	Son/daughter-in-law			0	0	181	6.5	353	6.7	34	11.6
	Parent			0	0	49	1.8	100	1.9	4	1.4
	Sibling			0	0	56	2	310	5.8	6	2.1
	Others			0	0	32	1.2	188	3.5	4	1.4
Experience of end-of-life discussion with patient	Yes			27	37.5						
	No			44	61.1						
Timing of patient’s death	Much sooner than expected			31	43.1						
	Sooner than expected			25	34.7						
	Moderate			9	12.5						
	Later than expected			5	6.9						
	Much later than expected			2	2.8						
Satisfaction with care											
on diagnosis	Satisfied			29	40.3						
	Not satisfied			43	59.7						
When patient became critical	Satisfied			31	38.9						
	Not satisfied			41	61.1						
When patient died	Satisfied			47	65.3						
	Not satisfied			25	34.7						
Financial impact of patient’s MPM on family	Significant impact			12	16.7						
	Some impact			15	20.8						
	Moderate impact			20	27.8						
	Minor impact			15	20.8						
	No impact			10	13.9						
Level of anger toward asbestos	Very angry			56	77.8						
	Angry			11	15.3						
	Moderately angry			4	5.6						
	Slightly angry			1	1.4						
	Not angry at all			0	0						
Age (in years)	Range:		32–82	Mean ± SD	62.5 ± 12.2	60.4 ± 12.5		59.3 ± 12.8		60.6 ± 12.1	
Time since bereavement (months)			9–110		45.2 ± 27.2	12.4 ± 3.5		11.8 ± 3.7		12.2 ± 6.6	

* Cited from the J-HOPE study (reference [21]). ** Pleural mesothelioma was classified as “Others” in the J-HOPE study. MPM = malignant pleural mesothelioma.

**Table 2 jcm-11-02541-t002:** Multiple regression model predicting good death (*n* = 72).

Dependent Variable: GDI Total Score (F = 9.098, *p* = 0.0001, Adjusted R^2^ = 0.260)						
Model	B	SE	β	t	95% CI	*p*-Value
Constant	41.724	4.769		8.794	32.202–51.246	0.001
Satisfied with care received when patient became critical	11.597	3.278	0.370	3.538	5.053–18.141	0.001
Female bereaved family member	11.061	4.028	0.284	2.746	3.018–19.103	0.008
Patient died later than expected	3.270	1.556	0.220	2.102	0.164–6.376	0.039

Abbreviations: F, overall F-test for regression; R^2^, correlation of determination; B, unstandardized coefficient; SE, standard error; β, standardized coefficient (beta); t, independent-sample *t* test; CI, confidence interval. Note: The variables included were as follows: patient’s age on diagnosis; sex of patient; survival; whether the patient received certified workmen’s accident compensation insurance; whether the patient was certified for asbestos-related health damage relief system; whether the patient received surgery; whether the patient received chemotherapy; whether the patient received palliative care; age of bereaved family member; sex of bereaved family member; timing of patient’s death; bereaved family members’ level of anger toward asbestos; the financial impact of the patient’s MPM on the family; whether bereaved family members were satisfied with the care received on diagnosis; whether bereaved family members were satisfied with the care received when the patient became critical; whether family members were satisfied with the care received at the point of death; the relationship of patient and bereaved family members; and whether family members had an end-of-life discussion with the patient.

**Table 3 jcm-11-02541-t003:** Multiple regression model predicting quality end-of-life care (*n* = 72).

Dependent Variable: CES Total Score (F = 34.558, *p* = 0.0001, Adjusted R^2^ = 0.493)						
Model	B	SE	β	t	95% CI	*p*-Value
Constant	30.545	1.807		16.907	26.939–34.152	0.001
Satisfied with the care received when the patient died	13.272	1.727	0.664	7.683	9.824–16.720	0.001
Received chemotherapy	4.048	1.832	0.191	2.209	0.391–7.705	0.031

Abbreviations: same as Table 2. Note: same as Table 2.

## Data Availability

The datasets generated and analyzed from this study are not publicly available to protect the anonymity of the participants but are available from the corresponding author, Yasuko Nagamatsu, upon reasonable request.

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
