# Peer review of "Bereaved Family Members’ Perspectives of Good Death and Quality of End-of-Life Care for Malignant Pleural Mesothelioma Patients: A Cross-Sectional Study"

_jcm, 2022, doi:10.3390/jcm11092541_

Round 1

Reviewer 1 Report

thank you very much for the opportunity to review this article
It is well written with the necessary details to understand the study. The methodology, results, and discussion are in agreement with the objectives.
I suggest that the authors better clarify how they obtained the results of the other study (are they from the same authors as this study?

Author Response

Response:

Thank you for your valuable comments and suggestion. Dr. Mitsunori Miyashita, one of our co-authors, conducted the research studies listed as References 21 and 24 and provided us the data. We added an explanation to clarify this point as follows (Page 3)

“2.4. Comparison of Study Data

A nationwide project to evaluate the hospice and palliative care in Japan was previously conducted by Miyashita et al. and reported as the Japan HOspice and Palliative care Evaluation (J-HOPE) study [21]. This project evaluated the end-of-life care of cancer patients from the perspective of bereaved family members in nationwide designated cancer centers, inpatient palliative care units (PCUs) and home hospices. The study focused on care satisfaction, the structure and process of care, and the achievement of a good death. This previous study compared the data according to the last place of care. Data from this previous study were provided to us by Dr. Miyashita who is a co-author of the present study. There were 8398 questionnaire responses from family members that were analyzed by Miyashita et al. [24].

Reviewer 2 Report

Dear Authors,

the study background, hypothesis and results are well presented and clear. Limitations are presented.

Some suggestions in order to enhance this study’s findings:

Please consider if some aspects regarding MPM (for istance, causes of this neoplasm) can be implemnted in the introduction and can help the argumentation in Conclusion.

Please in Methods section specify the outcome measures considered by authors (primary and potentially secondary outcomes).

Consider to explain in the paper the meaning of 6 months as exclusion criteria.

Please explain “Sex receiving treatments” (page 3).

Please in Conclusion summarise characteristics of a good death and a good quality of end-life care for MPM, suggested by these findings. Moreover, authors can add tips for further researches in this field.

Author Response

POINT-BY-POINT RESPONSES

Reviewer 2

Point 1: Please consider if some aspects regarding MPM (for instance, causes of this neoplasm) can be implemented in the introduction and can help the argumentation in Conclusion.

Response 1

We appreciate your helpful advice. We mentioned that MPM is caused by asbestos in the first sentence of the article. However, we are grateful for your suggestion to add a sentence showing how the patients feel death caused by asbestos in the Introduction section to help make a richer discussion and conclusion in the later parts of the article. We added the following sentence in the revised text (Pages 1): 

“1. Introduction

Malignant pleural mesothelioma (MPM) is a rare fatal malignancy mainly caused by asbestos [1]. The number of people with MPM who die each year in Japan is about 1550, and that number has been growing [2]. There are estimations that Japan will have 66,000–100,000 deaths from mesothelioma between the years 2003 and 2050 [3,4]. The median survival from the time of diagnosis in Japan is 7.9 months [5]. MPM causes a series of debilitating physical symptoms, such as chest pain, dyspnea, fatigue, anorexia, insomnia, constipation and sweating [6–11]. Psychological issues, such as uncertainty, lack of control [12], memory problems, difficulties in concentrating, feeling that problems cannot be solved [13], depression, anxiety, fear and isolation [8], all negatively affect the quality of life of MPM patients. Finally, there is additional psychological distress for victims of the asbestos industry [14]. Suffering from asbestos-related disease causes fear of premature death [15] .”

2:  Please in Methods section specify the outcome measures considered by authors (primary and potentially secondary outcomes).

Response 2:

Thank you for your important suggestion. We specified the primary outcomes and secondary outcomes we considered for this study in the Methods section as follows (Pages 2):

“2.2. Outcomes

Our primary outcomes were the achievement of a good death by MPM patients and quality of end-of-life care. The secondary outcome was the presence of the common symptoms of MPM.”

Point 3: Consider to explain in the paper the meaning of 6 months as exclusion criteria.

Response 3:

This research is part of a large study which also investigated the complicated grief of the bereaved family members of MPM. According to the previous study, diagnosis of complicated grief should be made at least 6 months later after death. We added an explanation to clarify this point as follows (Page 2):

“The exclusion criterion was a bereaved family member who had experienced a loss within six months. This research is part of a large study which also investigated the complicated grief of the bereaved family members of MPM patients. According to the previous study, the diagnosis of complicated grief should be made at least 6 months after the death of a family member [22].”

Point 4: Please explain “Sex receiving treatments” (page 4).

Response 4:

We apologize for the oversight. A comma was missing. We corrected the sentence as follows (Page 4):

“Thereafter, the GDI scores and the patients’ and bereaved participants’ information were examined. Sex, receiving treatments, approval for compensation, experience of end-of-life discussion with patients and satisfaction of care were treated as dichotomous variables.”

Point 5: Please in Conclusion summaries characteristics of a good death and a good quality of end-life care for MPM, suggested by these findings. Moreover, authors can add tips for further researches in this field.

Response 5:

We appreciate your helpful suggestions. This cross-sectional study revealed the remarkably rare achievement of a good death among MPM patients in Japan. The MPM patients experienced an enormous burden from symptoms and were seldom free of physical distress. We added a summary and points for future research as follows (Pages 12-13):

“Another challenge faced by MPM patients in the achievement of a good death was the sense of life completion, which was difficult for patients with MPM caused by asbestos. The quality of end-of life care for MPM patients was poorer than that of other cancer patients. The GDI score of the MPM patients was closely correlated with their CES score. Further research and interventions are urgently required, aimed at achieving a good death for MPM patients by providing quality continuous care, including (1) symptom management from the earliest stage; (2) care for psychological, social, and spiritual pain; and (3) care for their families as provided by a multidisciplinary team.”

Reviewer 3 Report

Thank you for this interesting submission. The paper explores the concept of good death but with specific patients; this is unique in attempting to avoid generalisations regarding the concept.

The paper is well-written and methodologically appropriate. A further read to iron out minor areas for readability purposes (minor proofreading) would benefit the text.

Apart from that, the paper is in a good standard for publication.

Author Response

Response:

We appreciate your favorable comments. The revised manuscript was proofread to ensure clarity and readability.